# Dependence of nucleosome mechanical stability on DNA mismatches

Thuy TM Ngo[1,2,3,4,5], Bailey Liu[6], Feng Wang[7], Aakash Basu[8,9], Carl Wu[10,11], Taekjip Ha[1,6,8,12,13,14]*

[1]Department of Physics, Center for Physics in Living Cells University of Illinois Urbana-Champaign, Urbana, United States; [2]Department of Molecular and Medical Genetics, Oregon Health and Science University, Portland, United States; [3]Cancer Early Detection Advanced Research Center (CEDAR), Knight Cancer Institute, Oregon Health and Science University, Portland, United States; [4]Department of Biomedical Engineering, Oregon Health and Science University, Portland, United States; [5]Division of Oncological Sciences, Oregon Health and Science University, Portland, United States; [6]Department of Biophysics, Johns Hopkins University, Baltimore, United States; [7]Laboratory of Biochemistry and Molecular Biology, Center for Cancer Research, National Cancer Institute, Bethesda, United States; [8]Department of Biophysics and Biophysical Chemistry, Johns Hopkins University, Baltimore, United States; [9]Department of Biosciences, Durham University, Durham, United Kingdom; [10]Department of Biology, Johns Hopkins University, Baltimore, United States; [11]Department of Molecular Biology and Genetics, Johns Hopkins University, Baltimore, United States; [12]Program in Cellular and Molecular Medicine, Boston Children's Hospital, Boston, United States; [13]Department of Pediatrics, Harvard Medical School, Boston, United States; [14]Howard Hughes Medical Institute, Boston, United States

*For correspondence:
taekjip.ha@childrens.harvard.edu

Competing interest: The authors declare that no competing interests exist.

**Abstract** The organization of nucleosomes into chromatin and their accessibility are shaped by local DNA mechanics. Conversely, nucleosome positions shape genetic variations, which may originate from mismatches during replication and chemical modification of DNA. To investigate how DNA mismatches affect the mechanical stability and the exposure of nucleosomal DNA, we used an optical trap combined with single-molecule FRET and a single-molecule FRET cyclization assay. We found that a single base-pair C-C mismatch enhances DNA bendability and nucleosome mechanical stability for the 601-nucleosome positioning sequence. An increase in force required for DNA unwrapping from the histone core is observed for single base-pair C-C mismatches placed at three tested positions: at the inner turn, at the outer turn, or at the junction of the inner and outer turn of the nucleosome. The results support a model where nucleosomal DNA accessibility is reduced by mismatches, potentially explaining the preferred accumulation of single-nucleotide substitutions in the nucleosome core and serving as the source of genetic variation during evolution and cancer progression. Mechanical stability of an intact nucleosome, that is mismatch-free, is also dependent on the species as we find that yeast nucleosomes are mechanically less stable and more symmetrical in the outer turn unwrapping compared to Xenopus nucleosomes.

## eLife assessment

This manuscript reports **important** data on the stability of nucleosomes with dsDNA substrates containing defined mismatches at three defined nucleosomal positions. **Compelling** evidence obtained by single-molecule FRET experiments shows that certain mismatches lead to more stable

nucleosomes likely because mismatches kink to enhance DNA flexibility leading to higher nucleosome stability. The biological significance and implications of the findings remain unclear.

## Introduction

DNA base-base mismatches are generated by nucleotide misincorporation during DNA synthesis, meiotic recombination, somatic recombination between nearly identical repeats, or chemical modification such as hydrolytic deamination of cytosine (*Li, 2008*). They are also introduced intentionally in some genome editing approaches (*Mertz et al., 2022*; *Rees and Liu, 2018*; *Chen and Liu, 2023*). DNA mismatches, if unrepaired, are sources of genetic variation such as single-nucleotide polymorphisms and point mutations which can alter the cellular phenotype and cause dysfunction, diseases, and cancer (*Li, 2008*; *Hanahan and Weinberg, 2011*). DNA mismatches also alter the physical properties of DNA such as local flexibility and conformational heterogeneity (*Fields et al., 2013*; *Isaacs and Spielmann, 2004*; *Wang et al., 2003*).

In eukaryotes, DNA is packaged into a basic unit, the nucleosome, which consists of 147 base pairs (bp) of DNA wrapped around a histone octamer core (*Kornberg, 1974*; *Chen et al., 2021*; *Luger et al., 2012*). In vivo, nucleosomes are regularly arranged along DNA like 'beads on a string', with short linker DNA separating the beads (*Kornberg, 1974*; *Chen et al., 2021*). It has been commonly observed that the rate of genetic variation along the genome is correlated with nucleosome positions. Although not without an exception (*Chen et al., 2012*), studies have shown that the base substitution rate is higher nearer the center of a nucleosome and increases with increasing nucleosome occupancy (*Semple and Taylor, 2009*; *Sasaki et al., 2009*; *Warnecke et al., 2008*; *Washietl et al., 2008*; *Sabarinathan et al., 2016*; *Tolstorukov et al., 2011*; *Hara et al., 2000*; *Yazdi et al., 2015*). One possible explanation for this correlation is that nucleosomes impose a barrier preventing the repair machinery from detecting and repairing a mismatch (*Li and Luscombe, 2020*) or a bulky DNA adduct induced by ultraviolet (*Sabarinathan et al., 2016*), thus leading to substitutions. Currently, it is unknown how substrates for DNA repair such as mismatches and bulky adducts may affect nucleosome mechanical stability and nucleosomal DNA unwrapping, which may affect accessibility of the nucleosomal DNA to the repair machinery.

RNA polymerase II can initiate transcription at 4 pN of hindering force (*Fazal et al., 2015*) and its elongation activity continues until it stalls at ~10 pN of hindering force (*Galburt et al., 2007*; *Schweikhard et al., 2014*). Therefore, the transcription machinery can generate picoNewtons (pN) of force on chromatin as long as both the machinery and the chromatin segment in contact are tethered to stationary objects in the nucleus. Another class of motor protein, chromatin remodeling enzymes, was also shown to induce processive and directional sliding of single nucleosomes when the DNA is under similar amount of tension (~5 pN; *Kim et al., 2024*). Therefore, measurements of nucleosomes at a few pN of force will expand our knowledge of the physiology roles of nucleosome structure and dynamics.

In an earlier work, we demonstrated a correlation between DNA flexibility and nucleosome stability under tension using the 601 nucleosome positioning sequence (*Ngo et al., 2015*). We showed that the 601 DNA around the histone core can unwrap asymmetrically under tension. One side of the outer DNA turn unwraps at a lower force and the other side unwraps at a higher force. The direction of asymmetry is controlled by the relative DNA flexibility of the two DNA halves flanking the dyad. Unwrapping force is lower for the nucleosomal DNA side with lower flexibility and vice versa. In addition, cytosine modifications that make DNA more flexible made the nucleosome mechanically more stable and vice versa (*Ngo et al., 2016*).

Here, we examined the effect of a DNA mismatch on DNA flexibility and nucleosome unwrapping dynamics. We used a single molecule DNA cyclization assay to examine the flexibility of DNA containing a mismatch, and a single-molecule fluorescence-force spectroscopy method to study the effect of mismatch on nucleosome unwrapping dynamics. We also examined the mechanical properties of nucleosomes assembled using yeast and *Xenopus* histones on intact DNA, that is no mismatch, in order to explore the effect of yeast specific histone features (*White et al., 2001*; *McBurney et al., 2016*) on nucleosome mechanical stability.

## Results

## Monitoring nucleosome unwrapping by fluorescence-force spectroscopy

To measure conformational dynamics of the nucleosome in response to external force we used a single-molecule assay that combines fluorescence resonance energy transfer (FRET) with optical tweezers (*Hohng et al., 2007*; *Zhou et al., 2011*; *Maffeo et al., 2014*). This assay allows us to use FRET to probe local conformational changes of the nucleosome caused by tension applied by optical tweezers through the two ends of the nucleosomal DNA.

The nucleosome was reconstituted using the nucleosome positioning sequence 601, with or without a C-C mismatch. We designed three DNA constructs 601-R18, 601-R39, and 601-R56 with the mismatches at R18, R39, and R56 positions situated in the middle of the outer turn, at the junction between the outer turn and inner turn, and in the middle of the inner turn, respectively (*Figure 1*). Because the distance between the mismatch positions (17, 21, and 38 bp) are not in multiples of 5 bp, they reside in different positions within their own super-helical turn. And the mismatches are at positions where the major groove face toward (R56) or away from (R18, R39) the histone core (R56) so they do not all share the same specific contacts with the histone octamer. Two fluorophores – Cy3 (FRET donor) and Cy5 (FRET acceptor) – were placed in appropriate positions to report on the unwrapping of various sections of nucleosomal DNA through reduction in FRET (*Figure 1* and *Figure 1—figure supplement 1*). The two strands of the DNA construct were separately created by ligation of the component strands (*Figure 1—figure supplement 1*) to ensure that the resulting DNA does not contain a nick. The double-stranded construct was then formed by slowly annealing the two purified ligated strands over 3–4 hr. All four DNA constructs (601, 601-R18, 601-R39, and 601-R56) yielded nucleosomes with the same electrophoretic mobility and single-molecule FRET value, indicating that the nucleosomes are homogeneously positioned for all four constructs (*Figure 1—figure supplement 1*). This is consistent with a previous single-nucleotide resolution mapping of dyad position from of a library of mismatches in all possible positions along the 601 sequence or a budding yeast native sequence which showed that a single mismatch (A-A or T-T) does not affect the nucleosome position (*Park et al., 2023*).

In the fluorescence-force spectroscopy assay, a nucleosome was anchored to a polymer-passivated glass surface via biotin-neutravidin linkage on one end of the nucleosomal DNA. The other end of the nucleosomal DNA was attached to a bead held in an optical trap via a $\lambda$-DNA tether (*Figure 1A*). As previously described (*Ngo et al., 2015*), we attached a pair of donor and acceptor fluorophores to the DNA to probe the unwrapping of nucleosomal DNA. To probe the unwrapping of the outer DNA turn, we constructed DNA with a labeling scheme called ED1 (end-dyad 1) in which the donor is incorporated on the 68th nucleotide from the 5' end of the top strand (I68) and the acceptor is attached to the 7th nucleotide from the 5' end of the bottom strand (J7) (*Figure 1A*). Upon nucleosome formation, the ED1 probe displayed high FRET due to proximity between the donor and the acceptor. We applied tension to the nucleosomal DNA by moving the piezo stage to which the glass surface attached at a constant speed of 455 nm/s while a focused laser (532 nm) follows the molecule to monitor fluorescence signals. The force increases nonlinearly and the loading rate, i.e. the rate at which the force increases, was approximately in the range of 0.2 pN/s to 6 pN/s, similar to the cellular loading rates for a mechanosensitive membrane receptor (*Jo et al., 2024*). The force was increased from a low value (typically between 0.4–1.0 pN) to a predetermined higher value and then returned to the low value by moving the stage in the opposite direction at the same speed (*Figure 1*). We observed a gradual decrease in FRET - corresponding to an increase in the Cy3-Cy5 distance - as the force increases. Upon further increase in force, we observed rapid fluctuations in FRET, followed by a sharp decrease in FRET (*Figure 1*), consistent with our previous studies (*Ngo et al., 2015*; *Ngo et al., 2016*) and a more recent study (*Díaz-Celis et al., 2022*) utilizing high resolution optical tweezers with simultaneous smFRET detection. Upon relaxation through gradual decrease in force, the nucleosome reformed as reported via recovery of high FRET but at a lower force than the force at which unwrapping occurred, demonstrating mechanical hysteresis.

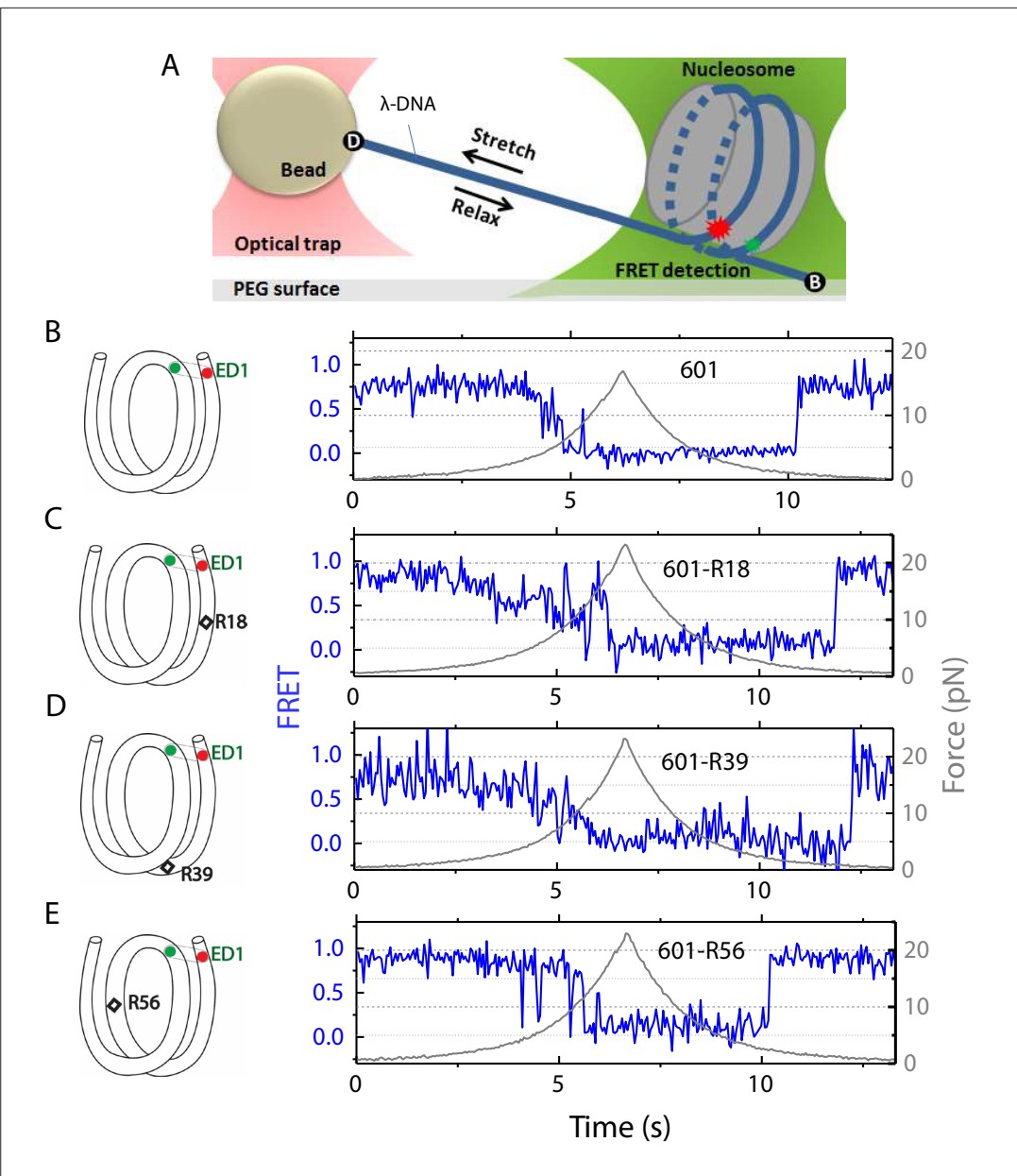

**Figure 1.** Nucleosome unwrapping measurement. (**A**) Experimental scheme. The red and green stars represent labelled Cy5 (acceptor) and Cy3 (donor) fluorophores, respectively. Biotin, B, and digoxigenin, D, are used to tether the nucleosome-lambda DNA construct to the surface and the bead, respectively. (**B, C, D, E**): Representative stretching traces of the outer turn (ED1) for nucleosomes reconstituted from the 601 sequence (**B**) and from the 601 sequence with containing a mismatch at different positions: on the outer turn (**C**), at the junction of the outer turn and inner turn (**D**) and at the inner turn (**E**). The red and green dots on the DNA bends represent labelled Cy5 and Cy3 fluorophores. The elongated circles enclosing red and green dots represent the ED labeling position. The black diamonds on the DNA bends represent the mismatch position with R18 and R39 on histone = facing minor grooves and R56 on a histone-facing major groove.

The online version of this article includes the following source data and figure supplement(s) for figure 1:

**Source data 1.** FRET efficiency and force vs time during force-induced unwrapping and rewrapping.

**Figure supplement 1.** Nucleosome preparation.

**Figure supplement 1—source data 1.** FRET efficiency distribution expressed as fraction per force bin.

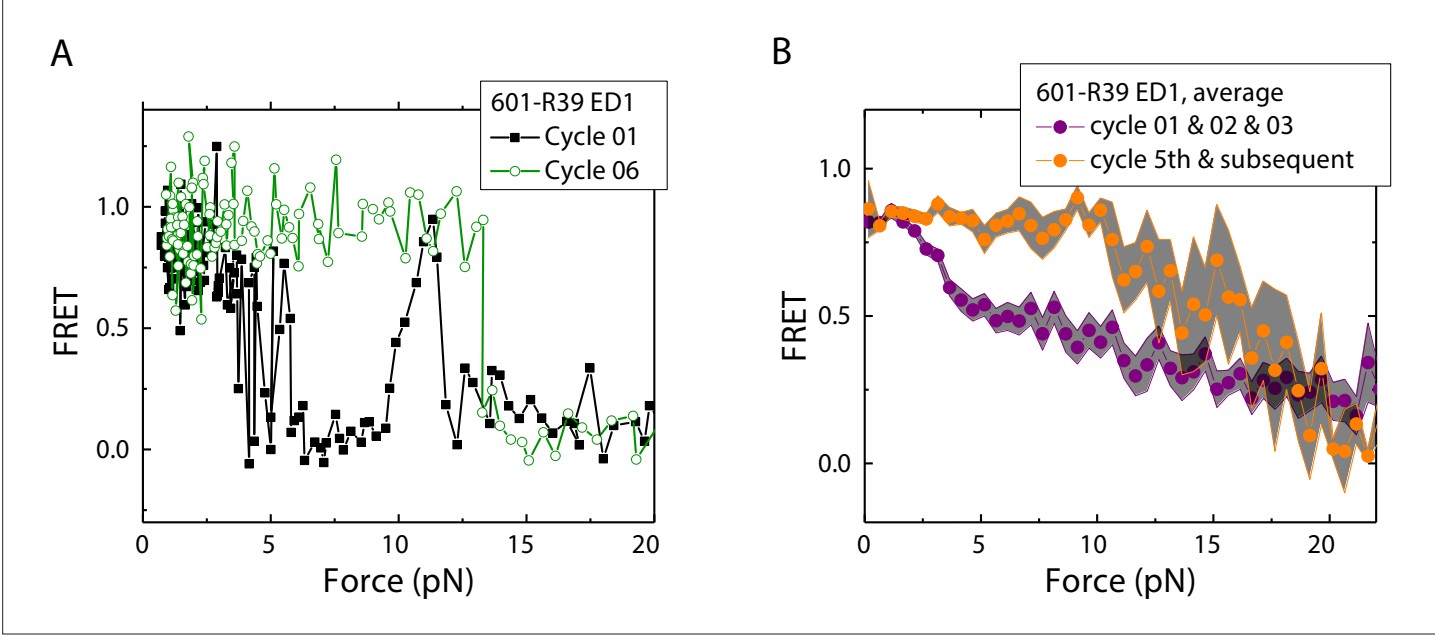

**Figure 2.** Unwrapping force of mismatch-containing nucleosomes is higher for subsequent stretching cycles. (**A**) Representative single-molecule stretching traces at two stretching cycles from the sample molecule, probe by the ED1 FRET pair in the 601-R18 nucleosome. (**B**) Averaging FRET vs. Force for many molecules at the first three stretching cycles (purple) and the subsequent stretching cycles (orange). Histone proteins were expressed in *Xenopus*. The error bars represent S.D. of n=25 and 11 traces for the first three stretching cycles (purple) and for the cycle 5th and the subsequent stretching cycles (orange), respectively.

The online version of this article includes the following source data for figure 2:

**Source data 1.** FRET efficiency vs Force during force-induced nucleosomal DNA unwrapping.

## History-dependent mechanical stability of mismatch-containing nucleosomes

Electrophoretic mobility shift analysis and zero-force FRET values did not show a noticeable difference between unmodified nucleosomes and mismatch-containing nucleosomes (*Figure 1—figure supplement 1*), consistent with a previous study that showed that, at single-nucleotide resolution, a single mismatch does not change the dyad position (*Park et al., 2023*). However, under perturbation by force, although unmodified nucleosomes showed the same behavior between stretching cycles (*Ngo et al., 2015*), mismatch-containing nucleosomes showed different behaviors between stretching cycles (*Figure 2*). The ED1 side of the mismatch containing nucleosomes unwrapped at lower forces for the first few cycles and then at higher forces for subsequent cycles. After relaxation, we observed a general trend of an increase in unwrapping force in the subsequent stretching cycles. One possible explanation for this observation is the re-positioning of the nucleosome such that the mismatch moves toward the dyad, bringing the ED1 probe toward the interior of the nucleosome, as predicted by a previous theoretical model (*LeGresley et al., 2014*). According to this model, the nucleosome position is weakly affected by the presence of a flexible lesion on the DNA, but under perturbation by other cellular components which either stiffen the DNA overall or weaken histone binding, the lesion can be made to have a strong preference for the dyad position. In our experiments, applied tension during stretching may act as perturbation which weakens nucleosome binding. When the probes move closer to the dyad in the subsequent stretching cycles, more base pairs of DNA would need to be unwrapped for FRET to decrease, potentially explaining the observed increase in unwrapping force. However, since the FRET values in our DNA construct are not sensitive to the nucleosome position, further experiments with fluorophores conjugated to strategic positions that allow discrimination between different dyad positions (*Blosser et al., 2009*) will be required to test this hypothesis.

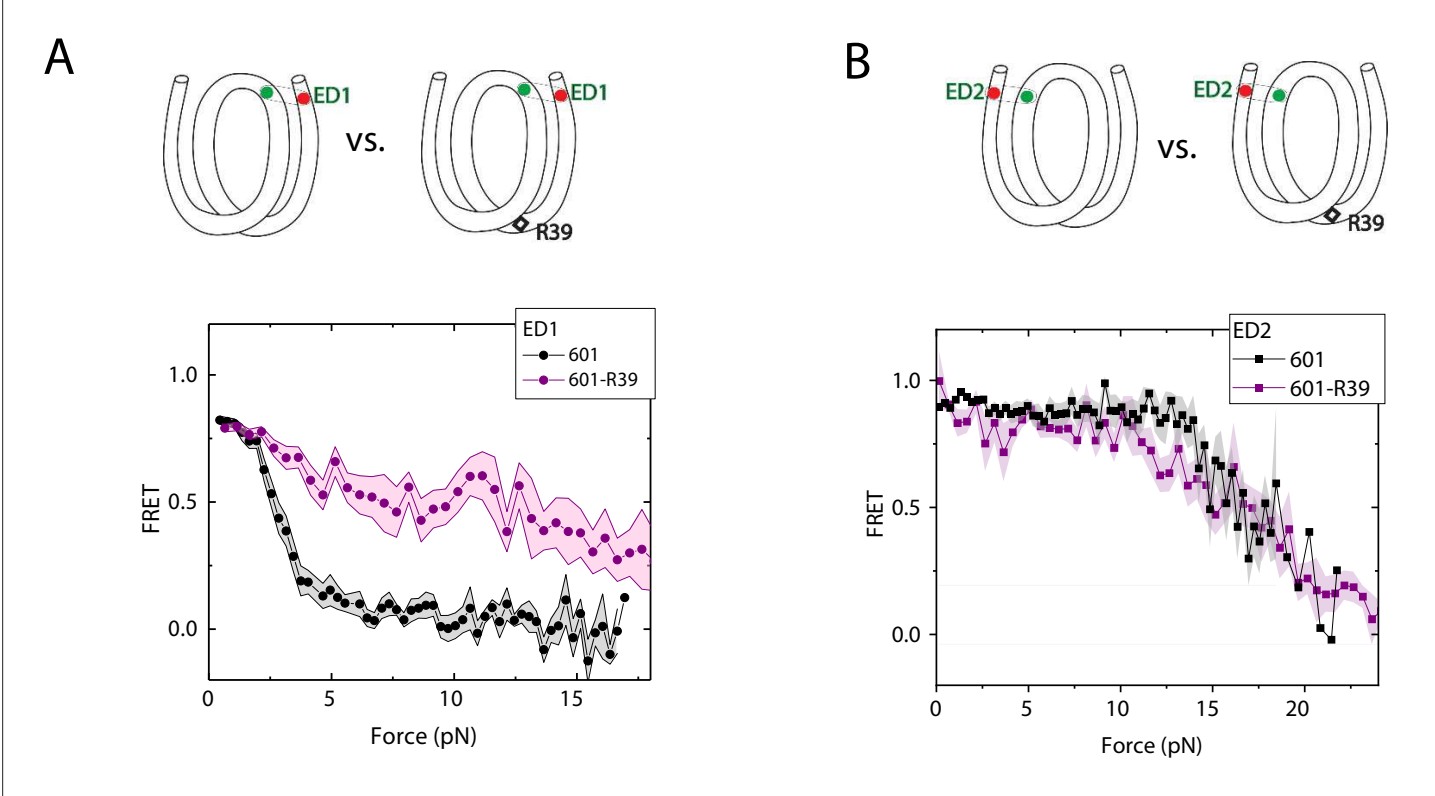

**Figure 3.** Enhancement of nucleosome mechanical stability by DNA mismatch. Average of FRET vs. Force for ED1 probe (**A**) and ED2 probe (**B**) for the 601 nucleosome (black) and for the first stretching cycle of the mismatch containing nucleosome 601-R39 (purple). Histone proteins were expressed in *Xenopus*. The error bars represent S.D. of n=25 and 7 for the ED1 probe of the 601 and 601-R39 nucleosomes (**A**) and n=20 and 39 for the ED2 probe of the 601 and 601-R39 nucleosomes (**B**), respectively.

The online version of this article includes the following source data for figure 3:

**Source data 1.** FRET efficiency vs force during force-induced DNA unwrapping from a nucleosome.

## DNA C-C mismatch enhances nucleosome mechanical stability

We compared the FRET versus force curves for constructs containing one C-C mismatch each at three different locations: R18, R39 and R56. We observed similar stretching patterns for the mismatch-containing nucleosomes (601-R18, 601-R39, 601-R56) to that of the 601 nucleosomes. However, the force range where FRET reduced gradually accompanied by fluctuations was wider and extended to higher force for mismatch-containing nucleosomes (*Figure 1C–E*). Because we observed increases in unwrapping forces for the second stretching cycle and beyond for mismatch-containing nucleosomes, we only used the first stretching cycle for comparing unwrapping forces between constructs. The averaged FRET vs. force pattern for 601-R39 showed an increase in unwrapping force for the mismatch-containing nucleosomes (*Figure 3A*). The increase in unwrapping force for all three mismatch containing constructs indicates that local flexibility of either the inner turn or the outer turn regulates nucleosome unwrapping (*Figure 4*). Next, we probed unwrapping of the nucleosome on the side that does not contain the mismatch for the construct containing a mismatch at the R39 position. In this configuration named ED2 (end-dyad 2), the donor was placed on the inner DNA turn close to the dyad (J58) which is similar to the ED1 construct, and the acceptor was incorporated to the opposite ends (I9) (*Figure 3B*). Stretching curves of ED2 nucleosomes formed on the 601 sequence yielded higher unwrapping force compared to the ED1 side as reported previously (*Ngo et al., 2015*). The mismatch construct yielded nearly the same unwrapping pattern as the 601 nucleosome, suggesting the change in local flexibility induced by the mismatch has a strengthening effect against unwrapping only for the side containing the mismatch (the ED1 side).

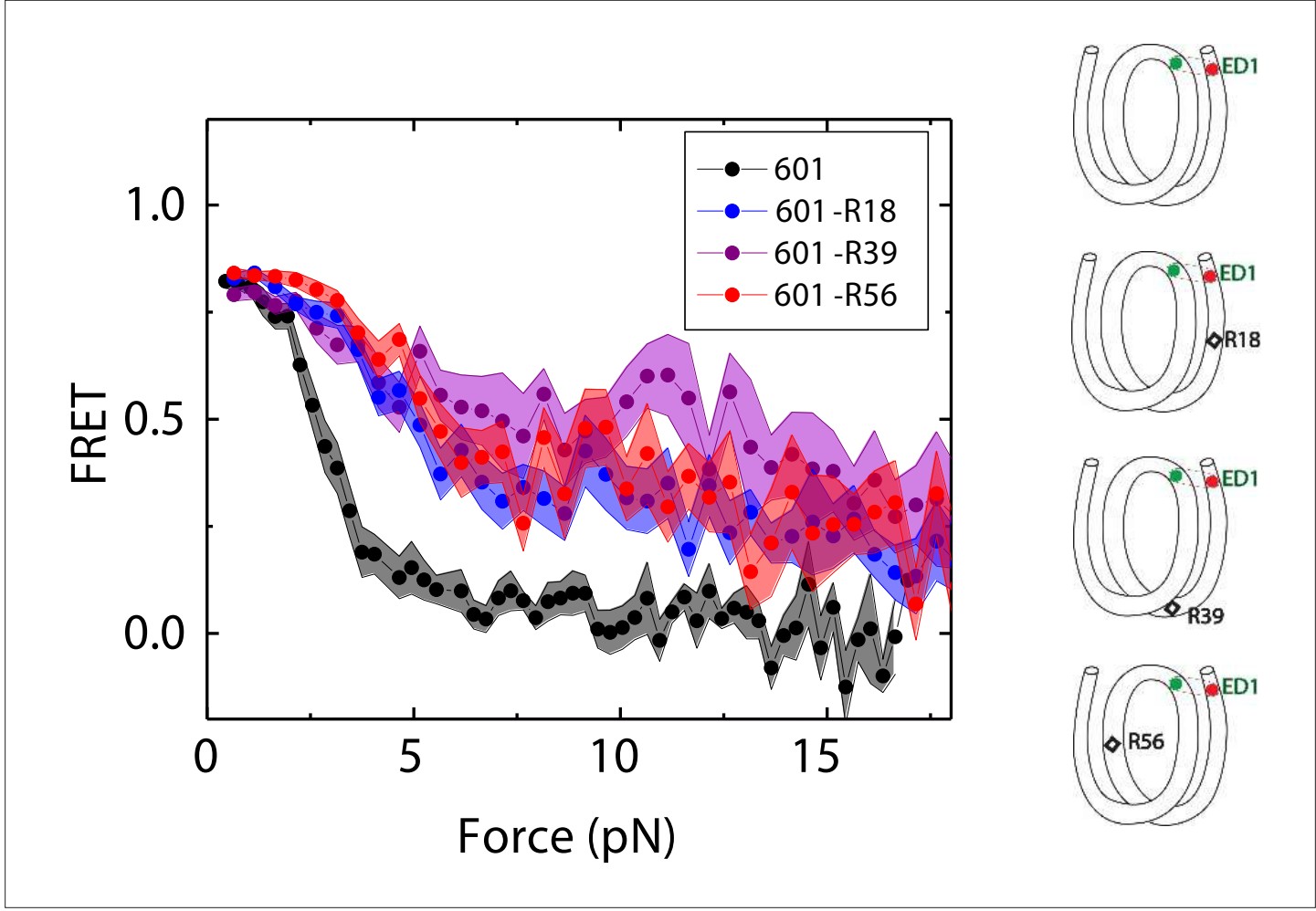

**Figure 4.** Mismatch position-dependence of nucleosome unwrapping. Average of FRET vs. Force for ED1 probe for the 601 nucleosome (black) and the mismatch-containing nucleosome 601-R39 (purple), 601-R18 (blue) and 601-R56 (red). Histone proteins were expressed in *Xenopus*. The error bars represent S.D. of n=25, 11, 7, and 10 for the 601, 601-R18, 601-R39, and 601-R56 nucleosomes, respectively.

The online version of this article includes the following source data for figure 4:

**Source data 1.** FRET efficiency vs force during force-induced DNA unwrapping from a nucleosome.

## DNA C-C mismatch enhances DNA bendability

A single DNA mismatch can cause DNA to deviate from the B-form conformation (*Isaacs and Spielmann, 2004*; *Wang et al., 2003*) and increase DNA flexibility (*Fields et al., 2013*). A previous study using a DNA buckling assay suggested that C-C is one of the most flexible mismatches (*Fields et al., 2013*). Therefore, we chose C-C as a representative mismatch to investigate its effect on nucleosome stability. We hypothesized that the stabilization of the nucleosome forming on mismatch containing DNA sequences is caused by its increase in DNA bendability. Therefore, we used a single-molecule DNA cyclization assay (*Vafabakhsh and Ha, 2012*) to probe the change in apparent bendability of the right half (RH) of the 601 sequence upon introducing the C-C mismatch. In this assay (*Figure 5* and *Figure 5—figure supplement 1*), DNA fragments with two 10 nt long 5' overhangs were immobilized on a microscope slide. A FRET pair (Cy3 and Cy5) was incorporated at the 5' ends of the overhangs that are complementary to each other, allowing us to detect high FRET when the two overhangs anneal with each other forming a circle. We used smFRET to quantify the fraction of looped molecules versus time after the high-salt buffer is introduced in the chamber. The rate of loop formation, which is the inverse of looping time determined from an exponential fitting of loop fraction vs time, was used as a measure of apparent DNA flexibility influenced by a mismatch (*Jeong et al., 2016*; *Jeong and Kim, 2019*). The faster the looping occurs, the more flexible the DNA is.

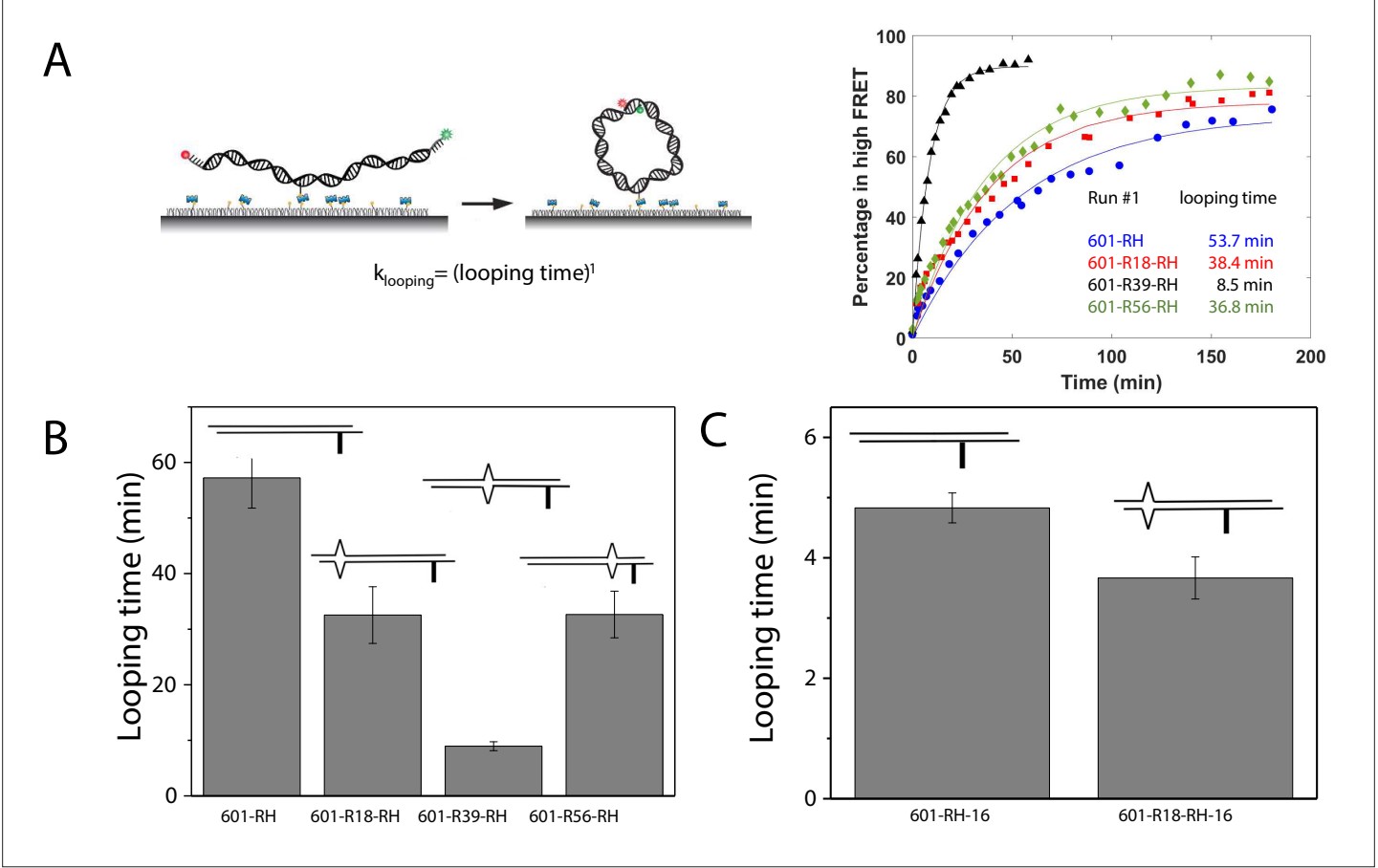

**Figure 5.** C-C mismatch enhances DNA flexibility. (**A**) Single-molecule cyclization assay: The DNA construct with 10-nucleotide complementary sticky ends is immobilized on a PEG passivated imaging chamber. DNA looping is induced using the imaging buffer containing 1 M NaCl followed by time course TIRF imaging. To calculate the looping time, the fraction of looped molecules (high FRET) as a function of time is fitted to an exponential function, $e^{-t/(looping\ time)}$ (right panel for one run of experiments). (**B, C**) Fitted looping time for the right half of the 601 construct without and with mismatches (**B**) and with the biotin position being moved by 16 nt (**C**). Error bars represented the S.E.M with n=3 technical replicates.

The online version of this article includes the following source data and figure supplement(s) for figure 5:

**Source data 1.** Quantification of single molecule looping kinetics.

**Figure supplement 1.** Single-molecule cyclization time course quantification.

**Figure supplement 1—source data 1.** High FRET fraction vs time for single molecule looping experiments.

We measured the looping time of 4 DNA constructs corresponding to the right half of the 601 sequence (601-RH) with the addition of a C-C mismatch at the R18, R39, and R56 locations (601-R18-RH, 601-R39-RH, and 601-R56-RH). As expected, we observed a dramatic decrease in looping time of the construct containing a mismatch (*Figure 5B*). Adding a C-C mismatch reduced the looping time from 57 min to 32 min (601-R18-RH), 9 min (601-R39-RH), and 32 min (601-R56-RH). The reduction in apparent looping time was larger with the mismatch placing at the center (601-R39-RH) than toward the side of the RH fragment (601-R18-RH,601-R56-RH) likely because the looping measurement is more sensitive to the change in flexibility at the center of the DNA fragment.

The cyclizability of surface-tethered DNA constructs was shown to possess an oscillatory dependence on the position of the biotin tether (*Basu et al., 2021*). For example, moving the location of the biotin tether by half the helical repeat (~5 bp) can lead to a large change in cyclization rate (*Basu et al., 2021*), likely due to the preferred poloidal angle of a given DNA (*Yoo et al., 2021*) that determines whether the biotin is facing towards the inside of the circularized DNA, thereby hindering cyclization due to steric hindrance caused by surface tethering. We therefore performed control experiments to test the possibility that the observed higher cyclization rates of constructs with mismatches, as shown in *Figure 5B–C*, is an artifact specific to the biotin tether location used. We

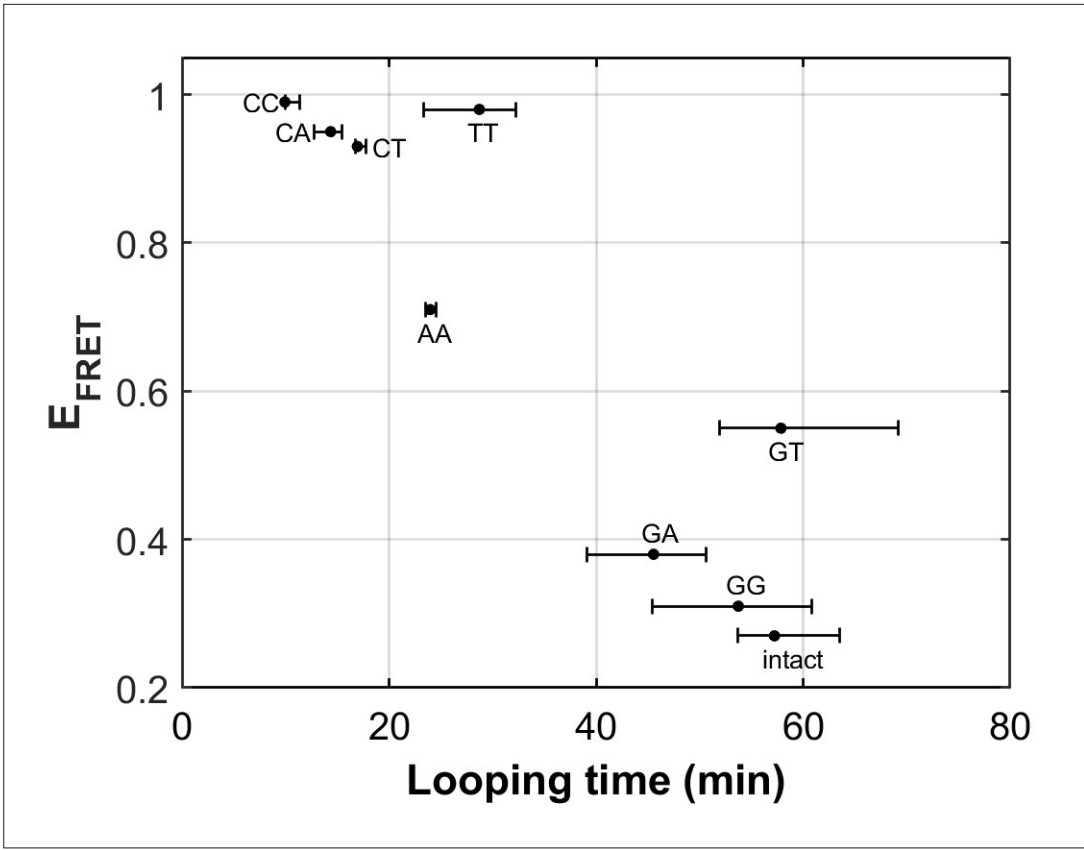

**Figure 6.** DNA flexibility enhancement is dependent on mismatch type. Looping times for DNA containing a single mismatch (one of eight types each) and an intact DNA without a mismatch. Also shown are ensemble FRET efficiencies ($E_{FRET}$) from **Fields et al., 2013** as a measure of DNA buckling for the same type of mismatch.

The online version of this article includes the following source data for figure 6:

**Source data 1.** FRET efficiency (a measure of DNA buckling) vs looping time for various mismatches.

created two additional constructs, 601-RH-16 and 601-R18-RH-16, which are identical to the 601-RH and 601-R18-RH constructs, respectively, except that the location of the biotin tether was moved to a thymine base that lies 16 nucleotides further towards the center of the molecule. We chose 16 nucleotides because it is about 1.5 times the helical repeat, and thus cyclization rates should be maximally different from those of the original 601-RH and 601-R18 constructs. Further, there was a thymine base present there to which the biotin could be conveniently attached. Side-by-side, we re-prepared the original 601-RH and 601-R18-RH constructs. We found that the overall looping rates of both the 601-RH-16 and 601-R18-RH-16 constructs were higher than those for the 601-RH and 601-R18-RH constructs, indicating that moving the biotin tether towards the center of the molecule increases looping rate (*Figure 5C*). However, the 601-R18-RH-16 construct, which contains a mismatch, still looped faster than the 601-RH-16 construct without a mismatch (*Figure 5C*). We thus conclude that the presence of the C-C mismatch makes the construct loop faster, and that this is not an artifact specific to the biotin tether location. As we will show next, the looping time for different mismatch types showed broadly similar behavior to that observed from DNA buckling experiments, further indicating that the mismatch effect is an intrinsic property.

## Effects of other mismatches on DNA bendability

There are eight different types of mismatches made from canonical DNA bases: A-A, T-T, C-C, G-G, G-A, C-A, C-T, and G-T mismatches. We performed single-molecule looping experiments of DNA containing a single mismatch introduced near the middle of the looping construct and determined the looping times for all eight constructs (*Figure 6*). See Supplementary Materials for their sequences. We observed significant reduction in looping time compared to the intact DNA control (no mismatch)

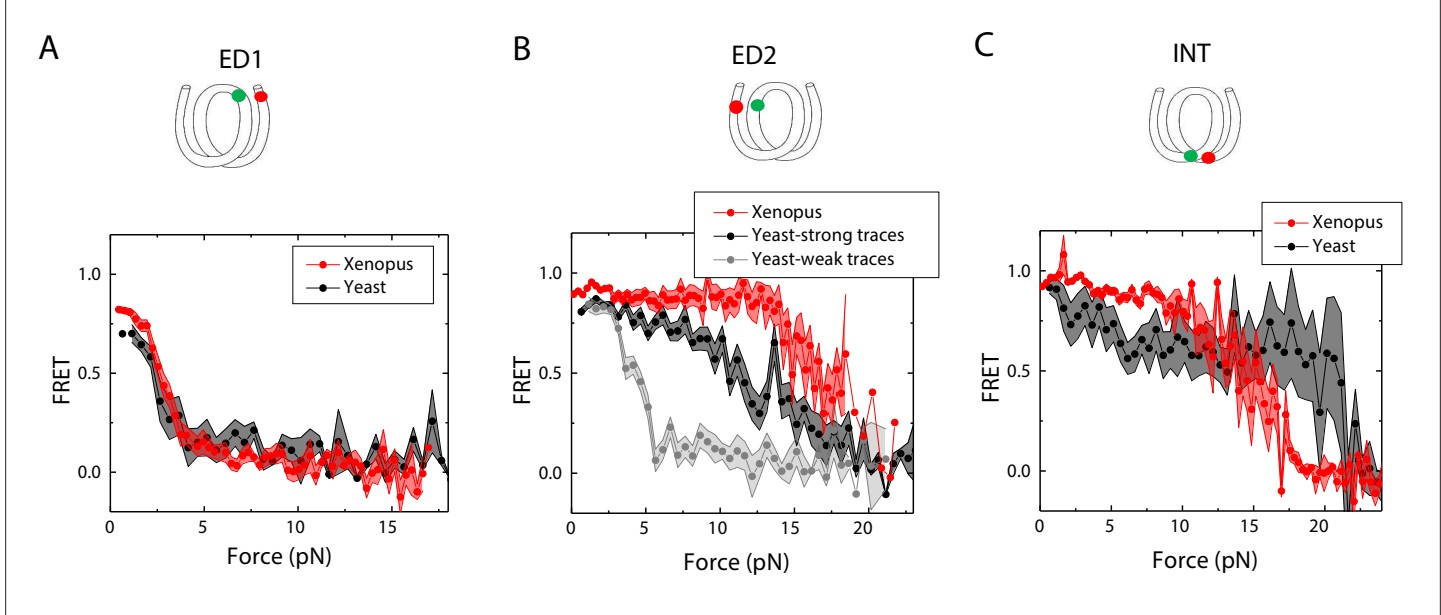

**Figure 7.** Unwrapping of yeast vs. *Xenopus* reconstituted nucleosomes. Average of FRET vs. Force for nucleosomes reconstituted from *Xenopus* (red) vs yeast (black and gray) histone proteins with DNA labeled by outer turn probes ED1 (**A**), ED2 (**B**) and inner turn probe INT (**C**). The error bars represent S.D. of n=17 (*Xenopus*) and 5 (Yeast) nucleosomes with the ED1 probe (**A**), n=20 (*Xenopus*), 6 (Yeast – strong) and 4 (Yeast-weak) nucleosomes with the ED2 probe (**B**), and n=22 (*Xenopus*) and 6 (Yeast) nucleosomes with the INT probe (**C**), respectively.

The online version of this article includes the following source data for figure 7:

**Source data 1.** FRET efficiency vs force during force-induced DNA unwrapping from a nucleosome.

with the exception of G-containing mismatches (G-G, G-T and G-A mismatches). The C-containing mismatches (C-C, C-A and C-T) showed the largest reduction in looping times, suggesting that they make DNA most bendable. We also compared our looping times with the published measure of DNA bendability for the corresponding mismatched DNA where they quantified DNA buckling via FRET ($E_{FRET}$ in *Figure 6*; *Fields et al., 2013*). The two measures generally agree. However, we found sizable deviations from a linear relation for T-T and G-T mismatches. It is possible that the deviations may arise from sequence contexts, but we cannot exclude the possibility that the two assays measure slightly different aspects of DNA mechanics. Although performing fluorescence-force spectroscopy for non-C-C mismatches is beyond the scope of the current work, a testable prediction is that T-T and A-A mismatches as well as C-containing mismatches make a nucleosome mechanically more stable.

## Yeast nucleosomes are less stable and more symmetrical than *Xenopus* nucleosomes in outer turn unwrapping

Next, we sought to examine how the source of histone proteins affects the mechanical stability of an intact nucleosome, that is without a mismatch. We reconstituted the 601 DNA construct with histone octamers of *Xenopus* and budding yeast. Note that all the data presented thus far on the effect of mismatches were obtained using *Xenopus* histones. Outer turn FRET probes on both sides ED1 and ED2 displayed slightly lower zero-force FRET values for yeast nucleosomes compared to *Xenopus* nucleosomes (*Figure 7*), indicating that the DNA entry/exit may be more loosely bound on histone core for yeast nucleosomes. In contrast, the inner turn FRET probe showed similar zero-force FRET values for yeast and *Xenopus* nucleosomes. With pulling force applied, the stretching pattern for ED1 is similar for both nucleosomes while the strong side probe ED2 showed lower mechanical stability for yeast histones, with 40% of the molecules having unwrapping force of lower than 5 pN and the other 60% of the molecules being unwrapped by a force between 5 and 15 pN (*Figure 7A–B*). The inner turn probe showed a stepwise unwrapping pattern with initial FRET reduction at less than 5 pN followed by stable FRET and a final unwrapping at a force higher than 20 pN (*Figure 7C*). These observations for both outer turn and inner turn probes suggested that nucleosomes made with yeast

histones are mechanically less stable, and unwrap less asymmetrically than nucleosomes made with *Xenopus* histones. Therefore, how DNA mechanics, determined by sequence, mismatch, or chemical modification, is translated to nucleosome mechanics may be influenced by the histone core.

## Discussion

Using the looping time of single-molecule DNA cyclization as a measure of DNA bendability, we showed that a DNA mismatch can increase DNA bendability. Our results for the selected mismatches are consistent with previous studies on the effect of a mismatch on DNA conformational dynamics using other methods such as NMR (*Isaacs and Spielmann, 2004*) and the DNA Euler buckling assay (*Fields et al., 2013*). A possible explanation for the enhancement of DNA flexibility is the existence of a kink at the mismatch position on the DNA. If and how the mismatch type-dependent DNA mechanics affects the sequence-dependent mismatch repair efficiency in vivo, as recently determined in a high through study in *E. coli* (*Kayikcioglu et al., 2023*), remains to be investigated. Comparison of mismatch-type dependent DNA mechanics to population genetics data is challenging because mutation profiles reflect a combined outcome of mismatch-generation, mismatch repair and selection in addition to other mutational processes.

We observed the enhancement of mechanical stability of nucleosomes reconstituted from mismatch-containing DNA constructs. A defect making the system more stable may appear counter-intuitive but given that the same mismatch can make DNA more flexible, our findings are in broad agreements with previous studies that showed positive correlation between DNA flexibility and nucleosome mechanical stability when DNA sequences or cytosine methylation was altered (*Ngo et al., 2015*; *Ngo et al., 2016*).

The 601 positioning sequence has TA-rich side that has four TA dinucleotides spaced with 10 bp periodicity and is more flexible than the TA-poor side (*Ngo et al., 2015*). The 601 nucleosome is more stable on the TA-rich side (*Ngo et al., 2015*; *Díaz-Celis et al., 2022*; *Tokuda et al., 2018*), which we attributed to the ease with which the more bendable DNA stays sharply bent around the histone core even under unwrapping force. Here, we introduced a mismatch to the TA-poor side with the aim of achieving a large contrast in the background of rigid DNA. Indeed, the mismatch induced a~ sevenfold increase in the rate of DNA cyclization of the TA-poor side. This increase in DNA flexibility matches the ~sevenfold larger cyclization rate of the TA-rich side compared to the TA-poor side (*Ngo et al., 2015*), suggesting that a single mismatch in the TA-poor side can symmetrize DNA flexibility of the 601 nucleosome. However, unlike flexibility symmetry achieved by TA repeats where which side unwraps at low forces became stochastic (*Ngo et al., 2015*), when the flexibility symmetry was obtained via a mismatch in the TA-poor side, the TA-rich side remained mechanically stable, unwrapping at only high forces. This difference suggests that although the apparent flexibility is similar between DNA containing a mismatch vs a flexible sequence element, the mismatch does not have a global effect on the coordination of unwrapping of the two DNA ends.

The enhanced nucleosome mechanical stability we observed suggests that a mismatch will reduce nucleosomal DNA accessibility. The reduction in nucleosomal DNA accessibility would hinder the activity of the DNA mismatch repair machinery on nucleosomal DNA. An unrepaired mismatch leads to a point mutation which may be the source for genetic variation during evolution and cancer progression. In fact, previous observations showed that the frequency of single-nucleotide polymorphism is higher near the nucleosome dyad for nucleosomes that are strongly positioned in vivo (*Li and Luscombe, 2020*). The higher frequency of substitutions in the nucleosomal DNA may be attributed to the difficulty of accessing the extra-stable nucleosomes. We also note that even without an enhanced stability, a mismatch within a nucleosome would be more difficult to detect for mismatch repair machineries compared to a mismatch in a non-nucleosomal DNA. Because mismatch repair machineries accompany the replisome, most of nascent mismatches may be detected for repair before nucleosome deposition. Therefore, the decrease in accessibility predicted based on our data here may be important only in rare cases a mismatch is not detected prior to the deposition of a nucleosome on the nascent DNA or in cases where a mismatch is generated via a non-replicative mechanism.

We chose the C-C mismatch for this work because a previous study showed that the C-C mismatch is one of the most flexible mismatches (*Fields et al., 2013*). If indeed more flexible elements in the DNA make a nucleosome mechanically stronger, as shown here for the C-C mismatch and previously for different sequences and cytosine modifications (*Ngo et al., 2015*; *Ngo et al., 2016*), we

can predict that other mismatches, DNA lesions and alternative DNA structures such as DNA single strand damages, bulky adducts and R-loops (*Huang and Zhou, 2021*) that alter DNA local flexibility would also change nucleosome mechanical stability accordingly. Future studies are needed to test this prediction.

We also tested if histones from different species and different histone variants can affect nucleosome stability under tension for an intact DNA without any mismatch. We observed a slightly lower zero-force FRET value for both sides with ED1 and ED2 for yeast nucleosomes compared to *Xenopus* nucleosomes. Under tension, we found that the outer turn of yeast nucleosomes could be unwrapped at a lower force than *Xenopus* nucleosomes, and that the unwrapping pattern is less asymmetric for ~40% of nucleosomes formed on the 601 sequence. The crystal structure of the yeast nucleosome suggests that yeast nucleosome architecture is subtly destabilized in comparison with nucleosomes from higher eukaryotes (*White et al., 2001*). Yeast histone protein sequences are not well conserved relative to vertebrate histones (H2A, 77%; H2B, 73%; H3, 90%; H4, 92% identities), and this divergence likely contributes to differences in nucleosome stability. Substitution of three residues in yeast H3 α3-helix (Q120, K121, K125) very near the nucleosome dyad with corresponding human H3.1/H3.3 residues (QK…K replaced with MP…Q) caused severe growth defects, elevated nuclease sensitivity, reduced nucleosome positioning and nucleosome relocation to preferred locations predicted by DNA sequence alone (*McBurney et al., 2016*). The yeast histone octamer harboring wild type H3 may be less capable of wrapping DNA over the histone core, leading to reduced resistance to the unwrapping force for the more flexible half of the 601positioning sequence. Overall, our data suggest that how DNA mechanics, determined by sequence (*Ngo et al., 2015*; *Basu et al., 2021*; *Basu et al., 2022*), chemical modifications (*Ngo et al., 2016*) and mismatches (*Vafabakhsh and Ha, 2012*; *Jeong and Kim, 2019*), is translated to nucleosome mechanics may be dependent on species-specific differences in histone sequence and post-translation modifications, which need to be examined in future studies.

Previous studies have showed that the artificial 601 sequence do not preferentially or strongly position the nucleosomes in vivo as expected (*Lancrey et al., 2022*; *Perales et al., 2011*). It is certainly desirable to perform mismatch-dependent and species-specific nucleosome mechanics studies using native sequences but the fluorescence-force spectroscopy data of the type we acquired in this study would be difficult to interpret unless the nucleosomes are formed at a well-defined position. We recently reported a native sequence from the yeast gene SWH1 that forms a nucleosome in vitro centered at the dyad position in vivo (*Park et al., 2023*).

While the enhancement of the mechanical stability of the nucleosome can potentially lead to the nucleosome's decreased accessibility for the mismatch repair mechanism, which can account for the accumulation of single-nucleotide polymorphisms near the nucleosome dyad, other functional implications of the enhanced mechanical stability cannot be precluded. For example, enhanced mechanical stability might shield DNA from transcription, which prevents the expression of genes that contain misincorporated nucleotides. Another opportunity for future studies is the fate of oligonucleosomes under tension when one of the nucleosomes contains a mismatch.

## Materials and methods
### Preparation of labeled DNA constructs

Each strand of DNA in constructs for cyclization measurements was prepared by ligation of two shorter DNA fragments containing labeled Cy3, Cy5 and biotin as indicated in *Figure 1—figure supplement 1*. Typically, the fragments were mixed at the ratio of 1:1.2:1.5 for the first, the helper, and the second fragments for ligation with T4 DNA ligase (NEB) following the manufacture manual. The ligation mixture was then loaded on a denaturing PAGE gel to run electrophoresis for purification. We cut and chop the top band which had the correct length and let the DNA diffuse to a buffer containing 10 mM Tris pH 8 and 50 mM NaCl. After purification, the two complementary strands were annealed by mixng at 1:1 molar ratio and heating to 90 °C followed by slow cooling over 3–4 hr. The final DNA construct contained the 601 sequence and was flanked by a 14 bp spacer to biotin for surface tethering and 20 bp spacer connect to a 12 nts overhang for annealing to lambda DNA.

## Nucleosome preparation

Both *Xenopus laevis* and yeast histones were expressed in *E. coli*. Yeast histones were prepared in C. Wu's lab at the National Institutes of Health as described (*Wang et al., 2016*). After purifying individual histone proteins, the histone octamers were prepared by denaturation-refolding and purification, according to standard procedures (*Dyer et al., 2004*). *Xenopus* histone octamers were purchased from The Histone Source, Colorado State University. To prepare nucleosomes, 601 DNA templates were reconstituted with the recombinant histone octamer by stepwise salt dialysis (*Dyer et al., 2004*). The reconstituted nucleosome product was confirmed by an electrophoresis mobility shift assay for all experiments. Reconstituted nucleosomes were stored at 4 °C in the dark, typically at concentrations of 100–200 nM, and used within 4 weeks.

## Single-molecule DNA cyclization measurement

DNA fragments for cyclization measurement were immobilized on a PEG-coated microscope slide via biotin-neutravidin linkage. The fragments had complementary 10 nt 5' overhang at either end, which permit looping via annealing. Cy3 and Cy5 were also present at the two 5' ends, resulting in high FRET in the looped state. Measuring FRET allowed us to quantify the fraction of looped molecules as a function of time since introduction of a high salt buffered solution 10 mM Tris-HCl pH 8.0, 1 M NaCl, 0.5% w/v D-Glucose (Sigma), 165 U/ml glucose oxidase (Sigma), 2170 U/ml catalase (Roche) and 3 mM Trolox (Sigma). Approximately 2500–3500 molecules were quantified at each timestamp during the experiment, and three independent experiments were performed for each sequence (*Figure 5—figure supplement 1*). The rate of loop formation was used as an operational measurement of DNA flexibility.

## Force-fluorescence spectroscopy measurement

We followed the protocol for force-fluorescence spectroscopy measurement published previously (*Ngo et al., 2015*; *Ngo et al., 2016*). To construct the DNA tether for a Force-Fluorescence measurement, $\lambda$ DNA was annealed to the reconstituted nucleosomes at one end, and to an oligonucleotide containing digoxigenin. The concentration of each element in the annealing reaction is 8 nM. During the experiment, the sample was diluted to 10 pM in nucleosome dilution buffer (10 mM Tris-HCl pH 8.0, 50 mM NaCl, 1 mM $MgCl_{2+}$ or 1 mM spermine) for immobilization on the PEG coated microscope slide. To attach the micro beads for optical trapping to the DNA construct. we diluted 1 μm anti-digoxigenin-coated polystyrene beads (Polysciences) in nucleosome dilution buffer and added it to the imaging chamber for 30 min. The fluorescence-force data acquisition procedures include three following steps using a custom built setup according to *Hohng et al., 2007*. First, after trapping a bead, we determined the origin of the tether by stretching it in two opposite directions along x and y axis. Second, to spatially avoid beaching of the fluorophores, we displaced the trapped bead from its origin where the labeled nucleosome is located by 14 μm. To locate the exact position of the label nucleosomes for confocal acquisition of the fluorescence signal, we scan the confocal laser around the tether's origin. Third, to apply the force on the tether, the nucleosome was stretched at a constant velocity of 455 nm/sec[26]. Fluorescence emission was recorded for 20ms at each step during the stage movement by scanning the confocal excitation concurrently with the stage movement. Force-fluorescence data was obtained in imaging buffer (50 mM Tris-HCl pH 8, 50 mM NaCl, 1 mM $MgCl_2$ or Spermine, 0.5 mg/ml BSA (NEB), 0.5 mg/ml tRNA (Ambion), 0.1% v/v Tween-20 (Sigma), 0.5% w/v D-Glucose (Sigma), 165 U/ml glucose oxidase (Sigma), 2170 U/ml catalase (Roche) and 3 mM Trolox (Sigma)). tRNA, which we normally include to reduce sticking of beads to the surface over the hours of single molecule experiments in a sealed chamber, was excluded in experiments with yeast-expressed nucleosomes because tRNA induced disassembly of nucleosomes assembled using yeast histones.

All single molecule measurements were performed at the room temperature.

## Acknowledgements

We thank Sergei Rudnizky for critical comments. This work was supported by the US National Institutes of Health (GM122569 to TH and GM125831 to CW) and by the National Science Foundation Physics Frontier Center program (PHY1430124). TTMN is supported by the Cancer Early Detection Advanced Research Center (CEDAR) at Oregon Health and Science University and grants from the

Department of Defense, Susan G Komen Foundation, and Kuni Foundation. FW was supported by the NCI intramural research program. CW was a NIH Scientist Emeritus and a Senior Fellow of the Howard Hughes Medical Institute Janelia Research Campus. AB is a Royal Society University Research Fellow. TH is an investigator with the Howard Hughes Medical Institute.

## Additional information

### Funding

| Funder | Grant reference number | Author |
|---|---|---|
| National Institute of General Medical Sciences | GM122569 | Taekjip Ha |
| National Institute of General Medical Sciences | GM132290 | Carl Wu |
| National Science Foundation | Physics Frontier Center program PHY1430124 | Taekjip Ha |
| National Cancer Institute | NCI intramural research program | Feng Wang |
| Howard Hughes Medical Institute Janelia Research Campus | | Carl Wu |
| Howard Hughes Medical Institute | | Taekjip Ha |

The funders had no role in study design, data collection and interpretation, or the decision to submit the work for publication.

### Author contributions

Thuy TM Ngo, Conceptualization, Formal analysis, Investigation, Methodology, Writing – original draft, Writing – review and editing; Bailey Liu, Investigation, Methodology, Writing – original draft, Writing – review and editing; Feng Wang, Investigation, Methodology; Aakash Basu, Investigation, Methodology, Writing – review and editing; Carl Wu, Funding acquisition, Methodology, Writing – review and editing; Taekjip Ha, Supervision, Funding acquisition, Writing – original draft, Project administration, Writing – review and editing

### Author ORCIDs

Bailey Liu https://orcid.org/0009-0004-5752-0119
Carl Wu http://orcid.org/0000-0001-6933-5763
Taekjip Ha http://orcid.org/0000-0003-2195-6258

Reviewer #1 (Public review): https://doi.org/10.7554/eLife.95514.3.sa1
Reviewer #2 (Public review): https://doi.org/10.7554/eLife.95514.3.sa2
Reviewer #3 (Public review): https://doi.org/10.7554/eLife.95514.3.sa3
Author response https://doi.org/10.7554/eLife.95514.3.sa4

## Additional files

### Supplementary files

• Supplementary file 1. Sequences for nucleosome reconstitution for the fleezers measurements – made by annealing the top and bottom strands which was constructed by ligation of two short fragments. Light gray shades denote sequences in the outer turn of the nucleosome and dark gray shares denote sequences in the inner turn of the nucleosome. Green and red 'T's denote the labeling sites for donor (Cy3) and acceptor (Cy5) fluorophores, conjugation done via amino-dT for the ED1 construct. For labeling sites for the ED2 construct, we refer to *Li, 2008*. Yellow highlights denote the positions of the mismatched bases. /idSp/ is the space added to prevent polymerization

onto the 5' overhang.

• MDAR checklist

## Data availability

Source data for all data figures in the manuscript have been provided as individual Microsoft Excel files.

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
