## [Editor Report · eLife assessment]

This manuscript reports **important** data on the stability of nucleosomes with dsDNA substrates containing defined mismatches at three defined nucleosomal positions. **Compelling** evidence obtained by single-molecule FRET experiments shows that certain mismatches lead to more stable nucleosomes likely because mismatches kink to enhance DNA flexibility leading to higher nucleosome stability. The biological significance and implications of the findings remain unclear.

---

## [Referee Report · Reviewer #1 (Public review)]

In this manuscript, Ngo et al. report a peculiar effect where a single base mismatch (CC) can enhance the mechanical stability of a nucleosome. In previous studies, the same group used a similar state-of-the-art fluorescence-force assay to study the unwrapping dynamics of 601-DNA from the nucleosome and observed that force-induced unwrapping happens more slowly for DNA that is more bendable because of changes in sequence or chemical modification. This manuscript appears to be a sequel to this line of projects, where the effect of CC is tested. The authors confirmed that CC is the most flexible mismatch using the FRET-based cyclization assay and found that unwrapping becomes slower when CC is introduced at three different positions in the 601 sequence. The CC mismatch only affects the local unwrapping dynamics of the outer turn of nucleosomal DNA.

---

## [Referee Report · Reviewer #2 (Public review)]

Mismatches occur as a result of DNA polymerase errors, chemical modification of nucleotides, during homologous recombination between near-identical partners, as well as during gene editing on chromosomal DNA. Under some circumstances, such mismatches may be incorporated into nucleosomes but their impact on nucleosome structure and stability is not known. The authors use the well-defined 601 nucleosome positioning sequence to assemble nucleosomes with histones on perfectly matched dsDNA as well as on ds DNA with defined mismatches at three nucleosomal positions. They use the R18, R39, and R56 positions situated in the middle of the outer turn, at the junction between the outer turn and inner turn, and in the middle of the inner turn, respectively. Most experiments are carried out with CC mismatches and *Xenopus* histones. Unwrapping of the outer DNA turn is monitored by single-molecule FRET in which the Cy3 donor is incorporated on the 68th nucleotide from the 5'-end of the top strand and the Cy5 acceptor is attached to the 7th nucleotide from the 5' end of the bottom strand. Force is applied to the nucleosomal DNA as FRET is monitored to assess nucleosome unwrapping. The results show that a CC mismatch enhances nucleosome mechanical stability. Interestingly, yeast and *Xenopus* histones show different behaviors in this assay. The authors use FRET to measure the cyclization of the dsDNA substrates to test the hypothesis that mismatches enhance the flexibility of the 601 dsDNA fragment and find that CC, CA, CT, TT, and AA mismatches decrease looping time, whereas GA, GG, and GT mismatches had little to no effect. These effects correlate with the results from DNA buckling assays reported by Euler's group (NAR 41, 2013) using the same mismatches as an orthogonal way to measure DNA kinking. The authors discuss that substitution rates are higher towards the middle of the nucleosome, suggesting that mismatches/DNA damage at this position are less accessible for repair, consistent with the nucleosome stability results.

---

## [Referee Report · Reviewer #3 (Public review)]

The mechanical properties of DNA wrapped in nucleosomes affect the stability of nucleosomes and may play a role in the regulation of DNA accessibility in eukaryotes. In this manuscript, Ngo and coworkers study how the stability of a nucleosome is affected by the introduction of a CC mismatched base pair, which has been reported to increase the flexibility of DNA. Previously, the group has used a sophisticated combination of single-molecule FRET and force spectroscopy with an optical trap to show that the more flexible half of a 601 DNA segment provides for more stable wrapping as compared to the other half. Here, it is confirmed with a single-molecule cyclization essay that the introduction of a CC mismatch increases the flexibility of a DNA fragment. Consistent with the previous interpretation, it also increased the unwrapping force for the half of the 601 segment in which the CC mismatch was introduced, as measured with single-molecule FRET and force spectroscopy. Enhanced stability was found up to 56 bp into the nucleosome. The intricate role of mechanical stability of nucleosomes was further investigated by comparing force-induced unwrapping profiles of yeast and *Xenopus* histones. Intriguingly, asymmetric unwrapping was more pronounced for yeast histones.

Note from Reviewing Editor:

The authors addressed the points in the reviews by making appropriate text additions and clarifications.

---

## [Author Response]

The following is the authors’ response to the original reviews.

**Public Reviews:**

**Reviewer #1 (Public Review):**
Summary:In this manuscript, Ngo et al. report a peculiar effect where a single base mismatch (CC) can enhance the mechanical stability of a nucleosome. In previous studies, the same group used a similar state-of-the-art fluorescence-force assay to study the unwrapping dynamics of 601-DNA from the nucleosome and observed that force-induced unwrapping happens more slowly for DNA that is more bendable because of changes in sequence or chemical modification. This manuscript appears to be a sequel to this line of projects, where the effect of CC is tested. The authors confirmed that CC is the most flexible mismatch using the FRET-based cyclization assay and found that unwrapping becomes slower when CC is introduced at three different positions in the 601 sequence. The CC mismatch only affects the local unwrapping dynamics of the outer turn of nucleosomal DNA.Strengths:These results are in good agreement with the previously established correlation between DNA bendability and nucleosome mechanical stability by the same group. This well-executed, technically sound, and well-written experimental study contains novel nucleosome unwrapping data specific to the CC mismatch and 601 sequence, the cyclizability of DNA containing all base pair mismatches, and the unwrapping of 601-DNA from xenophus and yeast histones. Overall, this work will be received with great interest by the biophysics community and is definitely worth attention.Weaknesses:The scope and impact of this study are somewhat limited due to the lack of sequence variation. Whether the conclusion from this study can be generalized to other sequences and other bendability-enhancing mismatches needs further investigation.Major questions:(1) As pointed out by the authors, the FRET signal is not sensitive to nucleosome position; therefore, the increasing unwrapping force in the presence of CC can be interpreted as the repositioning of the nucleosome upon perturbation. It is then also possible that CC-containing DNA is not positioned exactly the same as normal DNA from the start upon nucleosome assembly, leading to different unwrapping trajectories. What is the experimental evidence that supports identical positioning of the nucleosomes before the first stretch?

We added the following and refer to our recent publication1 to address this question.

“This is consistent with a previous single nucleotide resolution mapping of dyad position from of a library of mismatches in all possible positions along the 601 sequence or a budding yeast native sequence which showed that a single mismatch (A-A or T-T) does not affect the nucleosome position27.”

(2) The authors chose a constant stretching rate in this study. Can the authors provide a more detailed explanation or rationale for why this rate was chosen? At this rate, the authors found hysteresis, which indicates that stretching is faster than quasi-static. But it must have been slow and weak enough to allow for reversible unwrapping and wrapping of a CC-containing DNA stretch longer than one helical turn. Otherwise, such a strong effect of CC at a single location would not be seen. I am also curious about the biological relevance of the magnitude of the force. Can such force arise during nucleosome assembly in vivo?

To address the comment about the magnitude of force, we added the following paragraph to Introduction. “RNA polymerase II can initiate transcription at 4 pN of hindering force2 and its elongation activity continues until it stalls at ~ 10 pN of hindering force3,4. Therefore, the transcription machinery can generate picoNewtons of force on chromatin as long as both the machinery and the chromatin segment in contact are tethered to stationary objects in the nucleus. Another class of motor protein, chromatin remodeling enzymes, was also shown to induce processive and directional sliding of single nucleosomes when the DNA is under similar amount of tension (~ 5 pN)5. Therefore, measurements of nucleosomes at a few pN of force will expand our knowledge of the physiology roles of nucleosome structure and dynamics.”

To address the comment about the stretching rate, we added the following to Results. We note that the physiological loading rate has been challenging to determine for any biomolecular interactions, and the only quantitative measurement we are aware of is that of an integrin that we are citing.

“The force increases nonlinearly and the loading rate, i.e. the rate at which the force increases, was approximately in the range of 0.2 pN/s to 6 pN/s, similar to the cellular loading rates for a mechanosensitive membrane receptor6.”

(3) In this study, the CC mismatch is the only change made to the 601 sequence. For readers to truly appreciate its unique effect on unwrapping dynamics as a base pair defect, it would be nice to include the baseline effects of other minor changes to the sequence. For example, how robust is the unwrapping force or dynamics against a single-bp change (e.g., AT to GC) at the three chosen positions?

Unfortunately, we are unable to perform the suggested unwrapping experiment in a timely manner because the instrument has been disassembled during our recent move. However, we previously performed unwrapping experiments not only as a function of sequence but also as a function of cytosine modification and showed that we can detect even more subtle effects7,8. In addition, please note that we are not claiming that simply changing basepair at the chosen sites changes the mechanical stability of a nucleosome so we do not believe the requested experiment is necessary.

(4) The last section introduces yeast histones. Based on the theme of the paper, I was expecting to see how the effect of CC is or is not preserved with a different histone source. Instead, the experiment only focuses on differences in the unwrapping dynamics. Although the data presented are important, it is not clear how they fit or support the narrative of the paper without the effect of CC.

We apologize for giving the reviewer a wrong impression. We included the data because we believe that information on how the histone core can determine the translation of DNA mechanics into nucleosome mechanical stability will be of interest to the readers of this manuscript. We now mention explicitly that the observation was made using intact DNA, i.e. no mismatch, in the abstract and elsewhere.

(5) It is stated that tRNA was excluded in experiments with yeast-expressed nucleosomes. What is the reason for excluding it for yeast nucleosomes? Did the authors rule out the possibility that tRNA causes the measured difference between the two nucleosome types?

We normally include tRNA because we found that it reduces sticking of beads to the surface over several hours of experiments. In yeast nucleosomes, we found that tRNA causes the nucleosome to disassemble. Therefore, we did not include tRNA in yeast nucleosome experiments. We now mention this in Methods as reproduced below.

“tRNA, which we normally include to reduce sticking of beads to the surface over the hours of single molecule experiments in a sealed chamber, was excluded in experiments with yeastexpressed nucleosomes because tRNA induced disassembly of nucleosomes assembled using yeast histones.”

We cannot not formally rule out the possibility that tRNA causes the measured difference between *Xenopus* - vs Yeast- nucleosomes. However, we have shown in our previous publication7 that the asymmetric unwrapping in *Xenopus* nucleosomes was modulated by the DNA sequence. When we swapped the sequence of the inner turn between the two sides, while tRNA was included in all experiments, we observed stochastic unwrapping instead. As part of our response to another reviewer’s comments, we also added the following on the relevant differences between the species in Discussion.

“The crystal structure of the yeast nucleosome suggests that yeast nucleosome architecture is subtly destabilized in comparison with nucleosomes from higher eukaryotes9. Yeast histone protein sequences are not well conserved relative to vertebrate histones (H2A, 77%; H2B, 73%; H3, 90%; H4, 92% identities), and this divergence likely contributes to differences in nucleosome stability. Substitution of three residues in yeast H3 a3-helix (Q120, K121, K125) very near the nucleosome dyad with corresponding human H3.1/H3.3 residues (QK…K replaced with MP…Q) caused severe growth defects, elevated nuclease sensitivity, reduced nucleosome positioning and nucleosome relocation to preferred locations predicted by DNA sequence alone 10. The yeast histone octamer harboring wild type H3 may be less capable of wrapping DNA over the histone core, leading to reduced resistance to the unwrapping force for the more flexible half of the 601positioning sequence.”

**Reviewer #2 (Public Review):**
Summary:Mismatches occur as a result of DNA polymerase errors, chemical modification of nucleotides, during homologous recombination between near-identical partners, as well as during gene editing on chromosomal DNA. Under some circumstances, such mismatches may be incorporated into nucleosomes but their impact on nucleosome structure and stability is not known. The authors use the well-defined 601 nucleosome positioning sequence to assemble nucleosomes with histones on perfectly matched dsDNA as well as on ds DNA with defined mismatches at three nucleosomal positions. They use the R18, R39, and R56 positions situated in the middle of the outer turn, at the junction between the outer turn and inner turn, and in the middle of the inner turn, respectively. Most experiments are carried out with CC mismatches and *Xenopus* histones. Unwrapping of the outer DNA turn is monitored by singlemolecule FRET in which the Cy3 donor is incorporated on the 68th nucleotide from the 5'-end of the top strand and the Cy5 acceptor is attached to the 7th nucleotide from the 5' end of the bottom strand. Force is applied to the nucleosomal DNA as FRET is monitored to assess nucleosome unwrapping. The results show that a CC mismatch enhances nucleosome mechanical stability. Interestingly, yeast and *Xenopus* histones show different behaviors in this assay. The authors use FRET to measure the cyclization of the dsDNA substrates to test the hypothesis that mismatches enhance the flexibility of the 601 dsDNA fragment and find that CC, CA, CT, TT, and AA mismatches decrease looping time, whereas GA, GG, and GT mismatches had little to no effect. These effects correlate with the results from DNA buckling assays reported by Euler's group (NAR 41, 2013) using the same mismatches as an orthogonal way to measure DNA kinking. The authors discuss that substitution rates are higher towards the middle of the nucleosome, suggesting that mismatches/DNA damage at this position are less accessible for repair, consistent with the nucleosome stability results.Strengths:The single-molecule data show clear and consistent effects of mismatches on nucleosome stability and DNA persistence length.Weaknesses:It is unclear in the looping assay how the cyclization rate relates to the reporting looping time. The biological significance and implications such as the effect on mismatch repair or nucleosome remodelers remain untested. It is unclear whether the mutational pattern reflects the behavior of the different mismatches. Such a correlation could strengthen the argument that the observed effects are relevant for mutagenesis.
**Reviewer #3 (Public Review):**
Summary:The mechanical properties of DNA wrapped in nucleosomes affect the stability of nucleosomes and may play a role in the regulation of DNA accessibility in eukaryotes. In this manuscript, Ngo and coworkers study how the stability of a nucleosome is affected by the introduction of a CC mismatched base pair, which has been reported to increase the flexibility of DNA. Previously, the group has used a sophisticated combination of single-molecule FRET and force spectroscopy with an optical trap to show that the more flexible half of a 601 DNA segment provides for more stable wrapping as compared to the other half. Here, it is confirmed with a single-molecule cyclization essay that the introduction of a CC mismatch increases the flexibility of a DNA fragment. Consistent with the previous interpretation, it also increased the unwrapping force for the half of the 601 segment in which the CC mismatch was introduced, as measured with single-molecule FRET and force spectroscopy. Enhanced stability was found up to 56 bp into the nucleosome. The intricate role of mechanical stability of nucleosomes was further investigated by comparing force-induced unwrapping profiles of yeast and *Xenopus* histones. Intriguingly, asymmetric unwrapping was more pronounced for yeast histones.Strengths:(1) High-quality single-molecule data.(2) Novel mechanism, potentially explaining the increased prominence of mutations near the dyads of nucleosomes.(3) A clear mechanistic explanation of how mismatches affect nucleosome stability.Weaknesses:(1) Disconnect between mismatches in nucleosomes and measurements comparing *Xenopus* and yeast nucleosome stability.(2) Convoluted data in cyclization experiments concerning the phasing of mismatches and biotin site. ---
**Recommendations for the authors:**

**Reviewer #1 (Recommendations For The Authors):**
Specific comments:In Figure 1 legend, "the black diamonds on the DNA bends represent the mismatch position with R18 and R39 on minor grooves and R56 on a major groove." Minor and major grooves should be phrased as histone-facing minor and major grooves.

We fixed the problem.

In Materials and Methods, the sentence that describes the stretching rate cites reference 1, which does not seem to be relevant.

We fixed the problem.

**Reviewer #2 (Recommendations For The Authors):**
(1) In the introduction, the authors should also discuss the context of mismatches occurring during homologous recombination in meiosis or somatic cells in non-allelic recombination between near identical repeats.

Introduction now has the following.

“DNA base-base mismatches are generated by nucleotide misincorporation during DNA synthesis, meiotic recombination, somatic recombination between nearly identical repeats, or chemical modification such as hydrolytic deamination of cytosine.”

(2) Generally, it seems counter-intuitive in terms of biology that mismatches containing nucleosomes are more stable, as mismatches require repair and/or detection for heteroduplex rejection during recombination. Some discussion of this apparent paradox should be added.

To address this comment, we added the following to Discussion.

“The higher frequency of substitutions in the nucleosomal DNA may be attributed to the difficulty of accessing the extra-stable nucleosomes. We also note that even without an enhanced stability, a mismatch within a nucleosome would be more difficult to detect for mismatch repair machineries compared to a mismatch in a non-nucleosomal DNA. Because mismatch repair machineries accompany the replisome, most of nascent mismatches may be detected for repair before nucleosome deposition. Therefore, the decrease in accessibility predicted based on our data here may be important only in rare cases a mismatch is not detected prior to the deposition of a nucleosome on the nascent DNA or in cases where a mismatch is generated via a non-replicative mechanism.”

(3) The authors discuss that the substitution rate is higher while the indel (insertion and deletion) rate is lower nearer the center of a positioned nucleosome. Are the differences between individual mismatches reported in Figure 6 reflected in the mutagenic profile?

We cannot currently compare them because the mutagenic profile even when it is available is a complex convolution of mismatch generation, mismatch repair and selection. Mismatch generation occurs through several different processes and how they are affected by nucleosomes and their mismatch type and sequence context is unknown. Mismatch repair process itself depends on mismatch type and sequence context as recently shown by a high throughput in vivo study11. And because the population genetics does not simply reflect de novo mutation profiles due to selection, comparison between mismatch-induced DNA mechanical changes and mutagenic profiles is further complicated. We added the following to the revision.

“If and how the mismatch type-dependent DNA mechanics affects the sequence-dependent mismatch repair efficiency in vivo, as recently determined in a high through study in *E. coli* 11, remains to be investigated. Comparison of mismatch-type dependent DNA mechanics to population genetics data is challenging because mutation profiles reflect a combined outcome of mismatch-generation, mismatch repair and selection in addition to other mutational processes.”

(4) The looping assay should be explained better, especially how the cyclization rate is related to the reported looping time.

We modified Figure 5 to include examples of looping time determination through fitting of the looped fraction vs time, and added the following to the figure caption.

“To calculate the looping time, the fraction of looped molecules (high FRET) as a function of time is fitted to an exponential function, 𝑒−𝑡⁄(𝑙𝑜𝑜𝑝𝑖𝑛𝑔 𝑡𝑖𝑚𝑒) (right panel for one run of experiments).

Furthermore, we added the following sentence to Results.

“The rate of loop formation, which is the inverse of looping time determined from an exponential fitting of loop fraction vs time, was used as a measure of apparent DNA flexibility influenced by a mismatch 12,13.”

**Reviewer #3 (Recommendations For The Authors):**
I have some concerns that, when addressed upon revision, would improve the manuscript:(1) Page 6 and Supplementary Figure S1C: Though the FRET levels are the same for all nucleosomes, the distribution between the two levels is not. The nucleosomes with CC mismatches appear to have a larger fraction in the low-FRET population. This seems to contradict the higher mechanical stability. A comment on this should clarify it, or make this conundrum explicit.

Thank you for the comment. The low FRET population also includes the nucleosomes that do not have an active acceptor the fraction of which varies between preparations. We now note this in the supplementary figure caption.

(2) It is intriguing that a more stable nucleosome forms after several pulling cycles and it is argued that this might be due to shifting of the nucleosome. This seems reasonable and has important consequences both for the interpretation of the current experimental data and for the general mechanisms involved in nucleosome maintenance and remodeling. It is puzzling though how this would work mechanistically since it only seems to happen when nucleosomes are half-wrapped and when the unwrapped half contains the mismatch. From the previous work of the group and the current manuscript, it seems that shift does not occur in DNA without mismatches (Correct?). Does shifting happen for the 601-R18 and 601-R56 nucleosomes as well?

The mismatch-containing half is the half that is mechanically less stable in an intact, mismatch-free 601 nucleosome. So indeed, that is the half that is unwrapped in an intact nucleosome. But because the introduction of mismatch makes that half more mechanically stable, it can stay wrapped until higher forces, and the resulting structural distortion may cause the shift although we acknowledge that this interpretation remains speculative. Shifting occurs for all three constructs with a mismatch but not for the intact nucleosome without a mismatch.

(3) Could the shifting be related to the differences in sub-population distribution observed in Supplementary Figure S1C?

/See our response to comment (1) above.

(4) The paper would have more impact if the mechanism of possible shifting could be clarified. This can be done experimentally with a fluorescent histone, as suggested in the manuscript. But having a FRET pair on positions in the DNA that would shift to closer proximity upon shifting, either at the ED2 or at the ED1 site will also work, is in line with the current experiments and seems feasible.

We revised the text as follows in order not to exclude labeling configurations with both fluorophores on the DNA while reporting on the shift. We are also happy to add an appropriate reference if the reviewer can help us identify an existing study that measured dyad position shifts through such a labeling configuration.

“However, since the FRET values in our DNA construct are not sensitive to the nucleosome position, further experiments with fluorophores conjugated to strategic positions that allow discrimination between different dyad positions14 will be required to test this hypothesis.”

(5) Figures 5 and 6: To appreciate the quality of the data, state the number of molecules that contributed to the cyclization essay, or better, share a figure of the number of looped molecules as a function of time as supplementary data.

We added the requested figures to Figure 5 and a new supplementary Figure 2, and added the following to Methods.

“Approximately 2500 – 3500 molecules were quantified at each timestamp during the experiment, and three independent experiments were performed for each sequence (Supplemental Figure S2).”

(6) Page 8/9: A control is added to confirm that the phasing of the biotin relative to the end affects the observed cyclization rate. However, the mismatch sites were chosen such that they included 5 bp phase shifts. This convolutes the outcomes, as the direction of flexibility due to the phasing of the mismatch relative to the biotin may also influence the rate. Was this checked?

We would like to clarify that the phasing of the biotin is not so much as with respect to the end, as it is with respect to the full molecule. Static curvature and poloidal angle associated with the DNA molecule (which is something that is ultimately determined by the full chemical composition of the molecule, including its sequence and the mismatch) could make the molecule prefer a looped configuration where the biotin points towards the “inside” of the molecule. Such a configuration would be sterically unfavoured during the single molecule looping reaction where the biotin is attached to a surface via avidin. However, if the biotin is moved by half the helical repeat (or an off multiple of half the helical repeat, essentially 16 nt as done in the manuscript), it would now point to the “outside” of the molecule. Therefore, to make sure that the difference between the looping rates of any two DNA constructs (say the 601-RH and 601-R18-RH) is a better reflection of differences in dynamic flexibility, we ensure that the difference persists even when the biotin is moved by an odd multiple of half the helical repeat. We revised the section as follows.

“For example, moving the location of the biotin tether by half the helical repeat (~ 5 bp) can lead to a large change in cyclization rate15, likely due to the preferred poloidal angle of a given DNA16 that determines whether the biotin is facing towards the inside of the circularized DNA, thereby hindering cyclization due to steric hindrance caused by surface tethering.”

(7) Page 9/10: The comparison of yeast vs *Xenopus* is interesting, albeit a bit disconnected. Since the single-molecule statistics are relatively small, did the nucleosomes show similar bulk FRET distributions, or did they also show a shift in FRET levels?

We included the data because we believe that information on how the histone core can determine the translation of DNA mechanics into nucleosome mechanical stability will be of interest to the readers of this manuscript. The FRET values were similarly distributed.

(8) The discussion calls for a more detailed analysis of the structural differences of the histones of the two species to rationalize the observed asymmetry in flexibility dependence: why would yeast nucleosomes be less sensitive to sequence asymmetries?

We added the following to Discussion to address this comment.

“The crystal structure of the yeast nucleosome suggests that yeast nucleosome architecture is subtly destabilized in comparison with nucleosomes from higher eukaryotes9. Yeast histone protein sequences are not well conserved relative to vertebrate histones (H2A, 77%; H2B, 73%; H3, 90%; H4, 92% identities), and this divergence likely contributes to differences in nucleosome stability. Substitution of three residues in yeast H3 α3-helix (Q120, K121, K125) very near the nucleosome dyad with corresponding human H3.1/H3.3 residues (QK…K replaced with MP…Q) caused severe growth defects, elevated nuclease sensitivity, reduced nucleosome positioning and nucleosome relocation to preferred locations predicted by DNA sequence alone 10. The yeast histone octamer harboring wild type H3 may be less capable of wrapping DNA over the histone core, leading to reduced resistance to the unwrapping force for the more flexible half of the 601positioning sequence.”

(9) It would also be interesting if the increased stability due to the introduction of mismatches observed on *Xenopus* nucleosomes holds in yeast. Or does the reduced stability remove this effect? This is relevant to substantiate the broad claims in the context of evolution and cancer that are discussed in the manuscript.

Unfortunately, we are unable to perform the suggested unwrapping experiment in a timely manner because the instrument has been disassembled during our recent move. However, in terms of cancer relevance, our mismatch dependence experiments were performed using vertebrate nucleosomes (*Xenopus*) so repeating this for yeast nucleosomes would not provide relevant information.

Minor comments:(1) Supplementary Figure S1 misses the label '(C)' in its caption.

We fixed it.

(2) The supplementary data sequences for the fleezer measurements contain entrees 'R39 construct' and miss the positions of the Cy3 and Cy labels; the color code (levels of grey) is not explained.

We fixed the labeling mistake and added detailed annotations of the highlighted features.

References

(1) Park, S., Brandani, G.B., Ha, T. & Bowman, G.D. Bi-directional nucleosome sliding by the Chd1 chromatin remodeler integrates intrinsic sequence-dependent and ATP-dependent nucleosome positioning. Nucleic Acids Res 51, 10326-10343 (2023).

(2) Fazal, F.M., Meng, C.A., Murakami, K., Kornberg, R.D. & Block, S.M. Real-time observation of the initiation of RNA polymerase II transcription. Nature 525, 274-7 (2015).

(3) Galburt, E.A., Grill, S.W., Wiedmann, A., Lubkowska, L., Choy, J., Nogales, E., Kashlev, M. & Bustamante, C. Backtracking determines the force sensitivity of RNAP II in a factor-dependent manner. Nature 446, 820-3 (2007).

(4) Schweikhard, V., Meng, C., Murakami, K., Kaplan, C.D., Kornberg, R.D. & Block, S.M. Transcription factors TFIIF and TFIIS promote transcript elongation by RNA polymerase II by synergistic and independent mechanisms. Proc Natl Acad Sci U S A 111, 6642-7 (2014).

(5) Kim, J.M., Carcamo, C.C., Jazani, S., Xie, Z., Feng, X.A., Yamadi, M., Poyton, M., Holland, K.L., Grimm, J.B., Lavis, L.D., Ha, T. & Wu, C. Dynamic 1D Search and Processive Nucleosome Translocations by RSC and ISW2 Chromatin Remodelers. bioRxiv (2024).(6) Jo, M.H., Meneses, P., Yang, O., Carcamo, C.C., Pangeni, S. & Ha, T. Determination of singlemolecule loading rate during mechanotransduction in cell adhesion. Science (in press).

(7) Ngo, T.T., Zhang, Q., Zhou, R., Yodh, J.G. & Ha, T. Asymmetric unwrapping of nucleosomes under tension directed by DNA local flexibility. Cell 160, 1135-44 (2015).

(8) Ngo, T.T., Yoo, J., Dai, Q., Zhang, Q., He, C., Aksimentiev, A. & Ha, T. Effects of cytosine modifications on DNA flexibility and nucleosome mechanical stability. Nat Commun 7, 10813 (2016).

(9) White, C.L., Suto, R.K. & Luger, K. Structure of the yeast nucleosome core particle reveals fundamental changes in internucleosome interactions. EMBO J 20, 5207-18 (2001).

(10) McBurney, K.L., Leung, A., Choi, J.K., Martin, B.J., Irwin, N.A., Bartke, T., Nelson, C.J. & Howe, L.J. Divergent Residues Within Histone H3 Dictate a Unique Chromatin Structure in *Saccharomyces cerevisiae*. Genetics 202, 341-9 (2016).

(11) Kayikcioglu, T., Zarb, J.S., Lin, C.-T., Mohapatra, S., London, J.A., Hansen, K.D., Rishel, R. & Ha, T. Massively parallel single molecule tracking of sequence-dependent DNA mismatch repair in vivo. bioRxiv, 2023.01.08.523062 (2023).

(12) Jeong, J., Le, T.T. & Kim, H.D. Single-molecule fluorescence studies on DNA looping. Methods 105, 34-43 (2016).

(13) Jeong, J. & Kim, H.D. Base-Pair Mismatch Can Destabilize Small DNA Loops through Cooperative Kinking. Phys Rev Lett 122, 218101 (2019).

(14) Blosser, T.R., Yang, J.G., Stone, M.D., Narlikar, G.J. & Zhuang, X. Dynamics of nucleosome remodelling by individual ACF complexes. Nature 462, 1022-7 (2009).

(15) Basu, A., Bobrovnikov, D.G., Qureshi, Z., Kayikcioglu, T., Ngo, T.T.M., Ranjan, A., Eustermann, S., Cieza, B., Morgan, M.T., Hejna, M., Rube, H.T., Hopfner, K.P., Wolberger, C., Song, J.S. & Ha, T. Measuring DNA mechanics on the genome scale. Nature 589, 462-467 (2021).

(16) Yoo, J., Park, S., Maffeo, C., Ha, T. & Aksimentiev, A. DNA sequence and methylation prescribe the inside-out conformational dynamics and bending energetics of DNA minicircles. Nucleic Acids Res 49, 11459-11475 (2021).